# Enhanced ALOX12 Gene Expression Predicts Therapeutic Susceptibility to 5-Azacytidine in Patients with Myelodysplastic Syndromes

**DOI:** 10.3390/ijms25094583

**Published:** 2024-04-23

**Authors:** Taichi Matsumoto, Yuichi Murakami, Nao Yoshida-Sakai, Daisuke Katsuchi, Kuon Kanazawa, Takashi Okamura, Yutaka Imamura, Mayumi Ono, Michihiko Kuwano

**Affiliations:** 1Basic Medical Research Unit, St. Mary’s Research Center, 422, Tsubuku-Honmachi, Kurume 850-8543, Fukuoka, Japan; y-murakami@st-mary-med.or.jp (Y.M.); d-katsuchi@st-mary-med.or.jp (D.K.); ku-kanazawa@st-mary-med.or.jp (K.K.); mayumi512@aol.com (M.O.); mi-kuwano@st-mary-med.or.jp (M.K.); 2Department of Hematology, St. Mary’s Hospital, 422, Tsubuku-Honmachi, Kurume 850-8543, Fukuoka, Japan; nao-yoshida@st-mary-med.or.jp (N.Y.-S.); t-okamura@st-mary-med.or.jp (T.O.); y.imamura@st-mary-med.or.jp (Y.I.)

**Keywords:** myelodysplastic syndromes, 5-azacytidine, *ALOX12*

## Abstract

5-azacytidine (AZA), a representative DNA-demethylating drug, has been widely used to treat myelodysplastic syndromes (MDS). However, it remains unclear whether AZA’s DNA demethylation of any specific gene is correlated with clinical responses to AZA. In this study, we investigated genes that could contribute to the development of evidence-based epigenetic therapeutics with AZA. A DNA microarray identified that AZA specifically upregulated the expression of 438 genes in AZA-sensitive MDS-L cells but not in AZA-resistant counterpart MDS-L/CDA cells. Of these 438 genes, the *ALOX12* gene was hypermethylated in MDS-L cells but not in MDS-L/CDA cells. In addition, we further found that (1) the *ALOX12* gene was hypermethylated in patients with MDS compared to healthy controls; (2) MDS classes with excess blasts showed a relatively lower expression of *ALOX12* than other classes; (3) a lower expression of *ALOX12* correlated with higher bone marrow blasts and a shorter survival in patients with MDS; and (4) an increased *ALOX12* expression after AZA treatment was associated with a favorable response to AZA treatment. Taking these factors together, an enhanced expression of the *ALOX12* gene may predict favorable therapeutic responses to AZA therapy in MDS.

## 1. Introduction

Myelodysplastic syndromes (MDS) are a heterogeneous group of hematopoietic malignant disorders, in which various genetic and epigenetic aberrations occur in hematopoietic stem and progenitor cells [1,2]. MDS are characterized by cytopenia in single or multiple hematopoietic lineages, morphological abnormalities, ineffective hematopoiesis in the bone marrow, and a risk of progression to acute myeloid leukemia (AML) [3].

5-azacytidine (AZA) is a DNA-hypomethylating agent used as a conventional therapeutic for the treatment of high-risk MDS and AML with myelodysplasia-related changes (AML-MRC) [4,5]. AZA inhibits DNA methyltransferase 1 (DNMT1) by forming a covalent complex between DNMT1 and the DNA-azacytidine adduct, which is recognized by the ubiquitin–proteasome system and targeted for degradation, resulting in decreased DNA methylation and increased gene expression [6]. Approximately half of patients respond to AZA treatment, with improved overall survival and decreased leukemic transformation [7]. However, patients with MDS eventually relapse within two years, with poor outcomes [6,8], highlighting the need for a reliable strategy to identify patients who will benefit from AZA treatment. 

Concerning the identification of biomarkers predicting therapeutic responses to AZA in MDS, an earlier study reported that mutations in the Ten-Eleven Translocation 2 (*TET2*) gene impacted the outcome of AZA treatment; the response rate for AZA treatment was higher in patients with mutant *TET2* than wild-type TET2 [9]. However, other studies could not confirm the utility of the *TET2* mutation in predicting the response to AZA therapy [10,11,12]. A relevant study reported a close association between the *TP53* mutation and a response to AZA, but no significant association was found [13,14]. A superior response to AZA treatment was reported for patients with a mutation of DNA methyltransferase 3A (*DNMT3A*) in patients with AML [15,16]. However, whether the *DNMT3A* mutation is associated with a therapeutic response to AZA in MDS remains to be further studied.

In this study, we examined whether any gene could be specifically hypermethylated in patients with MDS and whether the methylation status of such a gene could play an essential role in chemotherapeutic responses to epigenetic therapeutics. We found that the expression of one hypermethylated gene, *ALOX12*, was highly susceptible to AZA treatment in bone marrow cells from patients with MDS and in MDS cells in vitro. We also discuss whether the methylation status of the *ALOX12* gene can predict the disease progression of MDS.

## 2. Results

### 2.1. Screening of AZA-Susceptible Genes

First, we screened genes that were highly susceptible to AZA. MDS-L and MDS-L/CDA cells (see Materials and Methods) were exposed to either 0 or 1 μM AZA for 72 h, and gene expression changes by AZA were comprehensively analyzed by a DNA microarray. We sorted genes that showed a specific increase in MDS-L cells but not in MDS-L/CDA cells (Figure 1A). AZA significantly increased the expression of 438 genes in MDS-L cells but not in MDS-L/CDA cells (Figure 1B). It has been previously reported that MDS are closely associated with abnormal differentiation [17] and apoptosis [18]. We focused on 27 differentiation- and 33 apoptosis-related genes among 438 genes (Figure 1B). AZA intensely (≥30-fold) increased the expression of the *ANGPT2*, *BIK*, *PRAME*, *CHAC1*, *ADCYAP1*, and *ALOX12* genes in MDS-L cells but not in MDS-L/CDA cells (Figure 1C). We validated the effect of AZA on the expression of these six genes in MDS-L and MDS-L/CDA cells by RT-qPCR. AZA dose-dependently increased the expression of all six genes in MDS-L cells (Figure 1D). By contrast, AZA did not augment the expression of the *ANGPT2*, *PRAME*, and *CHAC1* genes, and there was only a slightly, if any, augmented expression of the *BIK*, *ADCYAP1*, and *ALOX12* genes in MDS-L/CDA cells (Figure 1D).

### 2.2. Effect of AZA on the Methylation Status of the Six Selected Genes

We examined the effect of AZA on the methylation status of 5′-flanking regions of the above six genes in MDS-L and MDS-L/CDA cells by methylation-specific PCR (Figure 2A). AZA did not affect the methylation status of the 5′-flanking regions of *ANGPT2*, *BIK*, *PRAME*, *CHAC1*, and *ADCYAP1* in MDS-L and MDS-L/CDA cells (Figure 2B). By contrast, AZA specifically increased unmethylated DNA in the 5′-flanking regions (*ALOX12* R1 and R2) of the *ALOX12* gene in MDS-L cells but not in MDS-L/CDA cells (Figure 2B). Of the six genes, the 5′-flanking regions of the *ALOX12* gene were most frequently hypomethylated when treated with AZA, suggesting that the *ALOX12* gene is highly AZA-susceptible.

### 2.3. Methylation Status and mRNA Expression Levels of AZA-Susceptible Genes in Patients with MDS, Based on Public Dataset

We analyzed the methylation status of six AZA-susceptible genes in patients with MDS and healthy controls by retrieving the public dataset GSE152710. This dataset demonstrated methylation array data for bone marrow aspirates from 47 untreated patients with MDS and hematopoietic stem and progenitor fractions from 10 healthy controls. The methylation levels of the *ANGPT2*, *BIK*, *PRAME*, and *CHAC1* genes in patients with MDS were comparable to those in healthy controls. By contrast, the methylation levels of both the *ADCYAP1* and *ALOX12* genes were significantly higher in patients with MDS than in healthy controls (Figure 3A,B). 

Next, we analyzed expression levels of the *ADCYAP1* and *ALOX12* genes in patients with MDS and healthy controls by retrieving the public dataset GSE145733. This dataset showed microarray data for CD34^+^ bone marrow hematopoietic cells, including 54 untreated patients with MDS, 14 untreated patients with AML-MRC, and 9 healthy controls. In the World Health Organization classification 2017, MDS are classified into seven types: MDS with an isolated del(5q) [MDS-del(5q)], MDS with single-lineage dysplasia (MDS-SLD), MDS with multilineage dysplasia (MDS-MLD), MDS with ring sideroblasts and single-lineage dysplasia (MDS-RS-SLD), MDS with ring sideroblasts and multilineage dysplasia (MDS-RS-MLD), and MDS with excess blasts 1/2 (MDS-EB1/2) (Figure 3C,D). We compared the expression levels of *ALOX12* and *ADCYAP1* genes across seven MDS classes and AML-MRC. *ADCYAP1* expression levels were comparable between healthy controls, seven MDS classes, and AML-MRC (Figure 3C). By contrast, *ALOX12* expression levels were significantly lower in MDS-EB1, MDS-EB2, and AML-MRC than in other classes and healthy controls (Figure 3D). 

Since MDS-EB1, MDS-EB2, and AML-MRC are often accompanied by an increased number of bone marrow blasts [19], we examined whether expression levels of *ALOX12* were correlated with the proportion of bone marrow blasts in patients with MDS. The public dataset GSE58831 includes DNA microarray data of CD34^+^ bone marrow cells from 159 untreated patients with MDS and their number of bone marrow blasts and survival days. *ALOX12* expression levels were negatively correlated with the proportion of bone marrow blasts (Figure 3E). The overall survival of patients with a lower *ALOX12* gene expression tended to be shorter than that of patients with a higher *ALOX12* gene expression (Figure 3F). These results suggest that *ALOX12* expression influences the prognosis of patients with MDS by expanding bone marrow blasts.

### 2.4. Cell Growth and Lipid Peroxide Production in ALOX12-Overexpressing MDS-L Cells

*ALOX12* is a lipoxygenase that produces lipid peroxide, 12-hydroperoxieicosatetraenacid, from arachidonic acid [20], and *ALOX12* plays essential roles in reactive oxygen species-induced lipid peroxidation and cell death [21,22]. We examined whether *ALOX12* plays any role in cell growth and lipid peroxide production in MDS cells. We established *ALOX12*-overexpressing cells in MDS-L and MDS-L/CDA cells, respectively (Figure 4A). An overexpression of *ALOX12* had no effect on growth rates in both MDS-L and MDS-L/CDA cell lines (Figure 4B). We investigated the effect of tert-butyl hydrogen peroxide (TBH) on lipid peroxidation and cytotoxicity in MDS-L and MDS-L/CDA cells. TBH increased lipid peroxide 2.5-fold higher in MDS-L and MDS-L/CDA cells (Figure 4C). For both MDS-L and MDS-L/CDA cells, there was no difference in TBH-induced lipid peroxide production between controls (Empty) and ALOX12-overexpressing cells (ALOX12-OE) (Figure 4C). TBH dose-dependently suppressed cell survival by MDS-L and MDS-L/CDA cells at similar levels (Figure 4D), suggesting that it is less likely that *ALOX12* overexpression is directly involved in lipid peroxide production in MDS-L and MDS-L/CDA cells. *ALOX12* may indirectly contribute to blastogenesis suppression in patients with MDS, and how blastogenesis is inversely correlated with ALOX12 gene expression (see Figure 3E) should be further studied.

### 2.5. Expression of ALOX12 Gene during Treatment with AZA in Patients with MDS

Finally, we followed the expression levels of the *ALOX12* gene in bone marrow aspirates from nine patients with MDS, CMML, and AML-MRC before and after/during AZA treatment in our hospital (Figure 5A). *ALOX12* expression levels were not significantly different between CR (three cases), SD (three cases), and PD (three cases), before and after/during AZA treatment (Figure 5B,C). *ALOX12* expression levels increased in three of the six patients who showed favorable responses (CR and SD) to AZA treatment (#1, 2, and 4) and in one of the three patients with PD after AZA treatment (#7) (Figure 5D). Although the number of patients enrolled was small, an increased expression of ALOX12 mRNA by AZA treatment might likely be associated with a favorable response to treatment by AZA. More enrollment is required to clarify whether ALOX12 gene expression is associated with a therapeutic response to AZA in MDS.

## 3. Discussion

The development of evidence-based AZA therapeutics significantly contributes to the therapeutic improvement of patients with MDS. In our present study, we examined whether AZA’s demethylation of any specific gene could be associated with its therapeutic improvement of patients with MDS. We discovered that the expression of one gene, the *ALOX12* gene, was highly susceptible to AZA in vitro and that expression levels of *ALOX12* were closely correlated with disease progression in patients with MDS.

A comparison of genes upregulated by AZA between MDS-L and MDS-L/CDA cells identified six genes that were specifically upregulated by AZA in MDS-L cells (Figure 1). Of these six genes, the 5′-flanking regions of ALOX12 were strongly methylated and were specifically demethylated by AZA in MDS-L cells (Figure 2). These results suggested that the expression of the ALOX12 gene is suppressed by the methylation of its 5′-flanking regions and is susceptible to AZA treatment in MDS-L cells. Methylation levels of the *ALOX12* gene are markedly higher in MDS than in healthy controls (Figure 3B). Of the various MDS subclasses, *ALOX12* gene expression is lower in MDS classes with high bone marrow blasts than those with low bone marrow blasts. (Figure 3D), and expression levels of *ALOX12* gene are inversely associated with bone marrow blastogenesis (Figure 3E). A Kaplan–Meier analysis also showed a better prognosis for MDS with high *ALOX12* expression than MDS with low *ALOX12* expression (Figure 3F). The *ALOX12* gene may function as a suppressor of bone marrow blastogenesis and tumor progression in MDS. A previous study reported that the *ALOX12* gene was hypermethylated in 84% of patients with AML, including AML-MRC, and the hypermethylation of *ALOX12* had a worsening influence on the survival of patients with AML [23]. Although the effect of *ALOX12* methylation on the prognosis of patients with MDS is unknown to date, the methylation of the *ALOX12* gene might be associated with the pathogenesis and progression of MDS and AML. Further investigation is required to clarify the role of methylation in the *ALOX12* gene in MDS and AML.

It has been reported that *ALOX12* plays essential roles in reactive oxygen species-induced lipid peroxidation and cell death [21,22]. We investigated the effect of TBH on lipid peroxidation and cytotoxicity in MDS-L cells. However, *ALOX12* expression did not affect lipid peroxide production and cell death upon oxidative stress (Figure 4). These results suggest that *ALOX12* contributes to tumor suppression irrespective of the lipid peroxide production pathway in MDS. The *Alox15* gene, an ortholog of the human *ALOX12* gene, develops the myeloproliferative disease characterized by pancytopenia attributed to hematopoietic stem cell dysfunction at the asymptomatic stage, followed by an abnormal expansion of myeloid cells in the spleen and bone marrow [24,25]. *ALOX15* encodes arachidonate 12/15-lipoxygenase, which metabolizes arachidonic acid to 15(S)-hydroperoxy eicosatetraenoic acid (15(S)-HPETE) and 12(S)-hydroperoxy eicosatetraenoic acid (12(S)-HPETE) [26,27]. 12(S)-HPETE is a main metabolite of 12-lipoxygenase, a coding protein of the *ALOX12* gene. Ex vivo treatment with 12(S)-HPETE induces the cell death of aberrant hematopoietic progenitor cells from Alox15-deficient mice [24], suggesting the tumor-suppressive function of *ALOX12*. Further study should clarify how the *ALOX12* gene inhibits bone marrow blastogenesis in patients with MDS.

We compared expression levels of the *ALOX12* gene in bone marrow before and after or during AZA treatment in nine patients (#1–9), including CR (three cases), SD (three cases), and PD (three cases) (Figure 5A). Although there was no significant difference in the expression levels of the *ALOX12* gene between CR, SD, and PD, either before or after/during AZA treatment, the *ALOX12* gene expression levels were relatively higher in CR and SD than PD after AZA treatment (Figure 5B–D). Since this clinical study was performed with only nine cases, an increased number of patients should be further required to conclude whether ALOX12 is a valuable biomarker for developing epigenetic evidence-based therapeutics by AZA in MDS.

In conclusion, the present study raised the possibility that the suppression of ALOX12 gene expression through DNA methylation might promote the expansion of the BM blast, resulting in a poor prognosis. ALOX12 could be a promising candidate molecule for predicting the response to treatment with AZA in MDS.

## 4. Materials and Methods

### 4.1. Cell Culture

We previously established MDS-L cells resistant to AZA by transducing the cytosine deaminase (CDA), namely, MDS-L/CDA cells [28]. MDS-L cells were kindly provided by Dr. Kaoru Tohyama [29]. MDS-L and MDS-L/CDA cells were cultured in RPMI1640 medium (FUJIFILM Wako Pure Chemical Corporation, Osaka, Japan) supplemented with 10% fetal bovine serum (FBS) (Global Life Sciences Technologies Japan K.K., Tokyo, Japan) and 10 ng/mL recombinant human interleukine-3 (Thermo Fisher Scientific K.K., Tokyo, Japan). GP2-293 cells, a retrovirus-packaging cell line, were purchased from Takara Bio Inc. (Shiga, Japan) and cultured in DMEM (FUJIFILM Wako Pure Chemical Corporation) + 10% FBS. The cells were maintained in a humid atmosphere containing 5% CO_2_. 

### 4.2. DNA Microarray

MDS-L and MDS-L/CDA cells were treated with either 0.1% DMSO as a vehicle or 1 μM AZA (Selleck Biotech., Kanagawa, Japan) for 3 days. Total RNA was isolated using ISOGEN (Nippon Gene Co., Ltd., Tokyo, Japan) according to the manufacturer’s instructions. The DNA microarray and data analysis were performed by a Cell Innovator K.K. (Fukuoka, Japan). We defined the upregulation of gene expression as significant when the change ratio was ≥2 and the Z score was ≥2.

### 4.3. Reverse Transcription and Quantitative Polymerase Chain Reaction (RT-qPCR)

Total RNA was isolated from the cultured cells using ISOGEN reagent (Nippon Gene Co., Ltd.) according to the manufacturer’s instructions. The mixtures of primer pairs and probes (*ANGPT2*, Hs00169867_m1; *BIK*, Hs00154189_m1; *PRAME*, Hs01022301_m1; *CHAC1*, Hs00225520_m1; *ADCYAP1*, Hs00174950_m1; *ALOX12*, Hs00167524_m1; *RNA18S5*, Hs03928985_g1) were obtained from Thermo Fisher Scientific K.K. Reverse transcription and amplification were performed using an iTaq Universal One-Step reverse transcription and quantitative polymerase chain reaction (RT-qPCR) kit (Bio-Rad Laboratories, Inc., Tokyo, Japan), and gene amplification was monitored using a CFX96 Touch Real-Time PCR Detection System (Bio-Rad Laboratories, Inc.). The thermal cycling conditions included maintaining the reactions at 48 °C for 15 min and 95 °C for 10 min, followed by 40 cycles at 95 °C for 15 s and 60 °C for 1 min. *RNA18S5* was used as a reference gene, and the relative gene expression for each sample was determined using the ΔΔCt method. 

### 4.4. Methylation-Specific PCR (MSP)

Genomic DNA was extracted using GenElute Mammalian Genomic DNA (Merck KGaA, Darmstadt, Germany). The bisulfite treatment of the genomic DNA was performed using an EZ DNA Methylation Kit (Zymo Research Corporation, Tustin, CA, USA). A PCR was performed using an EpiScope^®^ MSP Kit (Takara Bio Inc., Shiga, Japan). We obtained the DNA sequences of the CpG island from the UCSC Genome Browser (https://genome-asia.ucsc.edu/index.html) accessed on 25 March 2024 and designed the primers for the MSP using Meth-Primer (https://www.urogene.org/cgi-bin/methprimer/methprimer.cgi) accessed on 25 March 2024. The sequences of primers for the MSP are listed in Appendix A. PCR products were subjected to 3% gel agarose electrophoresis. The gels were stained with Ethidium Bromide Solution (1:20,000, NACALAI TESQUE, Inc., Kyoto, Japan), and the images were captured using Amersham ImageQuant 800 (Global Life Sciences Technologies Japan K.K., Tokyo, Japan).

### 4.5. Establishment of ALOX12-Overexpressing Cells

The open reading frame of the *ALOX12* gene in 12 Lipoxygenase (ALOX12) Human Untagged Clone (OriGene Technologies, Inc., MD, USA), an ALOX12 expression plasmid, was amplified by PrimeSTAR Max DNA Polymerase (Takara Bio Inc.) according to the manufacturer’s instructions. The sequence of primers used for the amplification was as below: forward primer (5′→3′)—GCCG-CAGATCTATCGATGGGCCGCTACCGCATC, reverse primer (5′→3′)—TCGAC-GGATCCATCGTCAGATGGTGACACTGTTC. The PCR products were inserted into ClaI-digested pDON-AI-2 Neo DNA (Takara Bio Inc.), using an In-Fusion HD Cloning Kit (Takara Bio Inc.) according to the manufacturer’s instructions. Either pDON-AI-2 Neo DNA or *ALOX12* gene-inserted pDON-AI-2 Neo DNA and pVSV-G, a retrovirus envelopes-expressing plasmid, was co-transfected into GP2-293 cells with Lipofectamine LTX Reagent with PLUS Reagent (Thermo Fisher Scientific K.K.) according to the manufacturer’s instructions. After incubation at 37 °C for 3 days, the culture media were collected and filtered through a 0.45 mm filter. MDS-L and MDS-L/CDA cells were treated with a culture media of GP2-293 cells transfected with either Empty pDON-AI-2 Neo DNA or ALOX12-expression pDON-AI-2 Neo DNA and 8 μg/mL polybrene (NACALAI TESQUE, Inc., Kyoto, Japan) at 37 °C for 3 days. Cells transduced with Empty pDON-AI-2 Neo DNA or ALOX12-expression pDON-AI-2 Neo DNA were selected for treatment with 1 mg/mL G418 (Thermo Fisher Scientific K.K.) for 14 days.

### 4.6. Assessment of Doubling Time of Cells

Cells were seeded at 1 × 10^4^ cells/100 μL in the well of a 96-well plate and incubated at 37 °C in a 5% CO_2_ incubator. After 0, 24, 48, and 72 h, 7.5 μL of Cell Counting Kit-8 (Dojindo, Kumamoto, Japan) was added to the cell suspension. After incubation at 37 °C for 2 h, the absorbance of the culture medium at 450 nm was measured using an iMark plate reader (Bio-Rad Laboratories, Inc.). The doubling time of the cells was calculated using GraphPad Prism 10.2.0 (GraphPad Software Inc., San Diego, CA, USA).

### 4.7. Western Blotting

Western blotting was performed as described previously [28]. Cells were rinsed with ice-cold PBS and lysed in buffer containing 50 mM HEPES, 120 mM NaCl, 50 mM NaF, 1% Triton X-100, 10 glycerol, 5 mM EDTA, 1 mM phenylmethylsulfonyl fluoride, 10 μg/mL aprotinin, 10 μg/mL leupeptin, and 1 mM sodium orthovanadate. Cell lysates were separated by SDS-PAGE and transferred to an Immobilon-P PVDF Membrane (Merck KGaA, Darmstadt, Germany). After transfer, the membranes were incubated in a blocking solution and probed with anti-ALOX12 (ab211506, Abcam plc., Cambridge, UK) and anti-β-actin (ab8226, Abcam) antibodies. The luminescence intensity was quantified using an Amersham ImageQuant 800 (Global Life Sciences Technologies Japan K.K.).

### 4.8. Analysis of Gene Methylation Status in Patients with MDS

We utilized public datasets disclosed in the Gene Expression Omnibus (GEO). GEO accession number GSE152710 was used to analyze the methylation status of patients with MDS. The raw IDAT data and sample matrix sheet were downloaded and analyzed by shinyÉPICo [30], a graphical pipeline to analyze Illumina DNA methylation arrays, installed into R for Windows version 4.2.2.

### 4.9. Detection of Intracellular Lipid Peroxide

Cells were treated with 0 or 200 μM tert-butyl hydrogen peroxide (TBH) (FUJIFILM Wako Pure Chemical Corporation) for 2 h at 37 °C and were then stained with 5 μM BODIPY581/591 C11 (C11-BODIPY) (Thermo Fisher Scientific K.K.) for 30 min at 37 °C. After the cells were washed with PBS twice, the fluorescent intensity of C11-BODIPY was measured using BD FACSCalibar (BD, Franklin Lakes, NJ, USA).

### 4.10. Patients and Tissue Samples

This study involved nine patients treated with AZA: six with MDS (five with RAEB-1 and one with RAEB-2), one with CMML, and two with AML with myelodysplasia-related changes (AML-MRC), who were diagnosed between 2012 and 2021 at the Department of Pathology of St. Mary’s Hospital. This study was conducted according to the guidelines of the Declaration of Helsinki and approved by the Institutional Review Board of St. Mary’s Hospital (15-0606) on 6 June 2015. Informed consent was obtained from all the subjects involved in this study. The clinical and laboratory information of the patients were obtained from their medical records. The International Working Group (IWG) response criteria were used to classify the therapeutic responses to AZA as complete remission (CR), stable disease (SD), or progressive disease (PD). Frozen bone marrow aspirates harvested from patients before and after AZA therapy were used for mRNA expression analysis using qRT-PCR.

### 4.11. Statistical Analysis

Statistical analysis was performed using GraphPad Prism 10.2.0 (GraphPad Software Inc.). Statistical significance was set at *p* < 0.05.

## Figures and Tables

**Figure 1 ijms-25-04583-f001:**
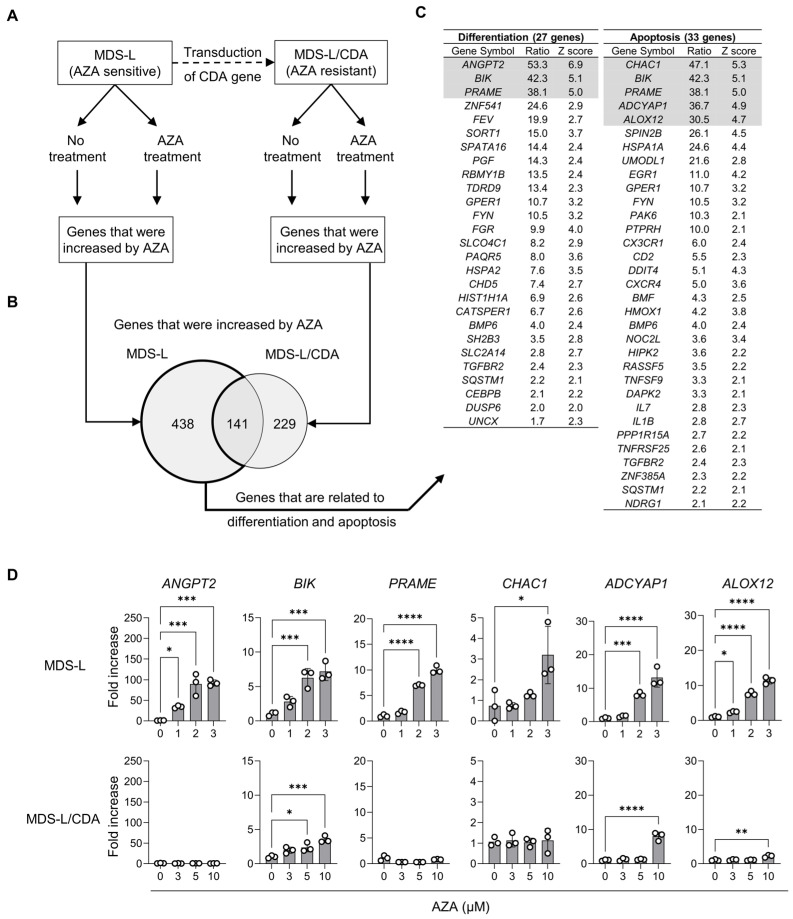
Selection of AZA-susceptible genes in MDS. (**A**) Schema for selection of AZA-susceptible genes in MDS. MDS-L, and MDS-L/CDA cells that were treated with 0 or 1 μM AZA for 3 days; we compared AZA-induced gene expression between MDS-L and MDS-L/CDA cells by DNA microarray. (**B**) Venn diagram shows increased genes in MDS-L and MDS-L/CDA cells treated with AZA. Expression of 438 genes was specifically increased in MDS-L cells when treated with AZA. (**C**) The list of genes related to differentiation and apoptosis among 438 genes. The gray column indicates genes more than 30-fold increased by AZA treatment in MDS-L cells. (**D**) The effect of AZA treatment on mRNA expression of *ANGPT2*, *BIK*, *PRAME*, *CHAC1*, *ADCYAP1*, and *ALOX12* genes in MDS-L and MDS-L/CDA cells. MDS-L and MDS-L/CDA cells were treated with 0, 1, 2, and 3 or 0, 3, 5, and 10 μM AZA for 72 h, respectively, and mRNA expression was measured by RT-qPCR. Experiments were repeated three times, and data are expressed as the mean ± standard deviation (SD). Significant differences were analyzed using a one-way analysis of variance, followed by Dunnett’s multiple comparisons test. **** *p* < 0.001, *** *p* < 0.005, ** *p* < 0.01, and * *p* < 0.05.

**Figure 2 ijms-25-04583-f002:**
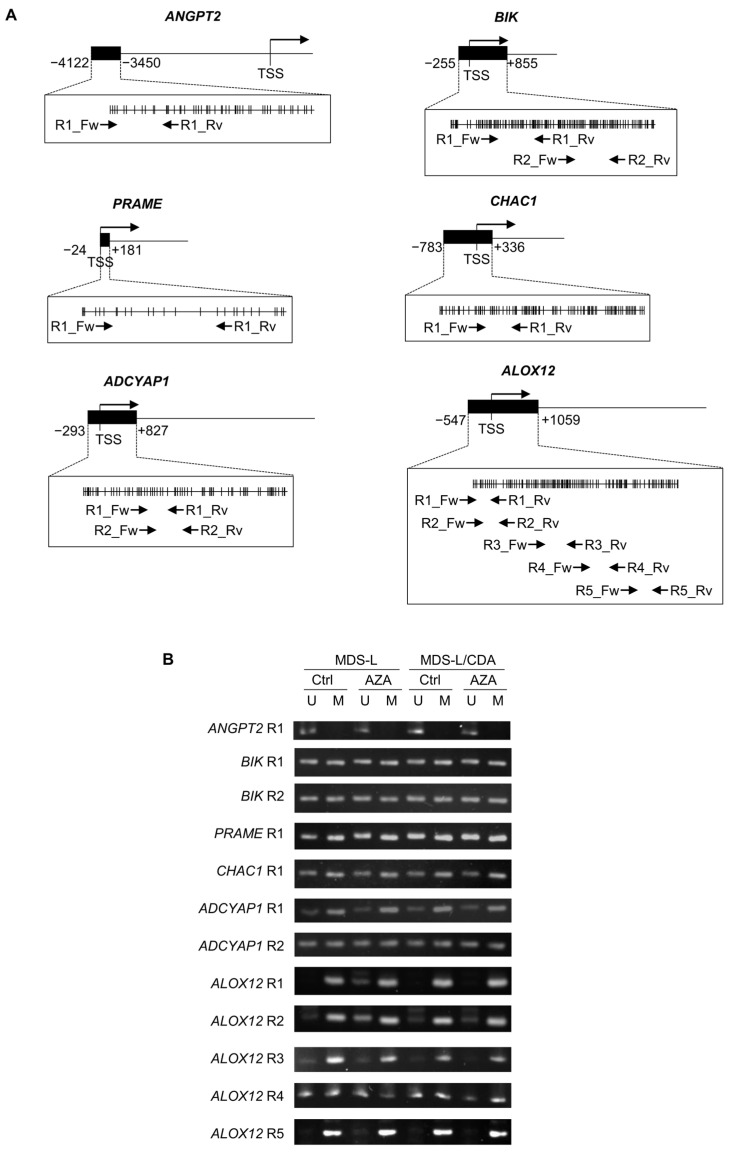
Effect of AZA on methylation status in 5′-flanking regions of *ANGPT2*, *BIK*, *PRAME*, *CHAC1*, *ADCYAP1*, and *ALOX12* genes. (**A**) Analyzed regions in the CpG island of ANGPT2, BIK, PRAME, CHAC1, ADCYAP1, and ALOX12 genes for methylation-specific PCR. ■; CpG island, TSS; transcription start site, |; CpG site, R; region, Fw; forward primer, Rv; reverse primer. (**B**) Agarose gel electrophoresis of PCR product by methylation-specific PCR. MDS-L and MDS-L/CDA cells were treated with 0 or 1 μM AZA for 72 h, and their genomic DNA was subjected to bisulfite treatment and methylation-specific PCR. U; unmethylated, M; methylated.

**Figure 3 ijms-25-04583-f003:**
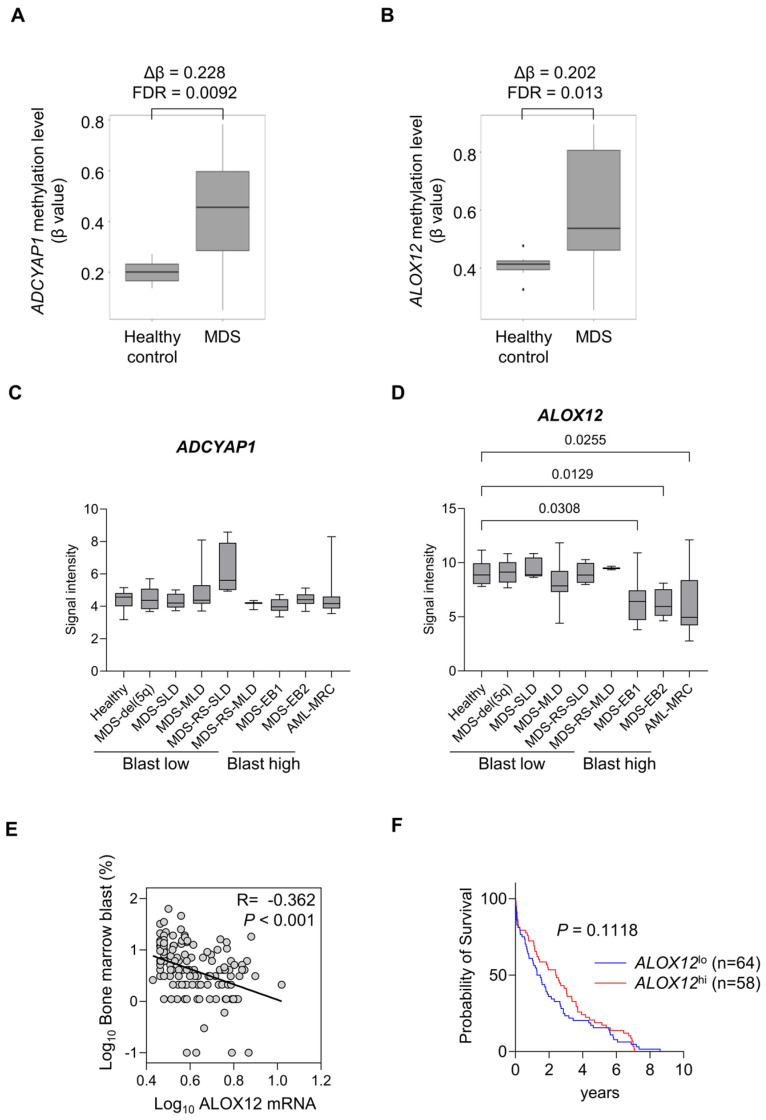
Methylation status in 5′-flanking regions of *ALOX12* genes in patients with MDS. (**A**,**B**) Methylation levels of each CpG island in 5′-flanking regions of *ADCYAP1* (**A**) and *ALOX12* (**B**) genes in healthy controls and patients with MDS, based on analysis of public dataset GSE152710. Δβ; β_MDS_ − β_healthy_, FDR; false discovery rate. (**C**,**D**) Expression levels of *ADCYAP1* (**C**) and *ALOX12* (**D**) in various MDS classes and healthy controls and AML-MRC patients. The significance of differences in gene expression was analyzed using the Kruskal–Wallis test. *p* value was indicated if significant difference was found. Data were obtained from public dataset GSE145733. (**E**) Correlation mRNA expression levels of *ALOX12* genes with proportion of bone marrow (BM) blasts. (**F**) Patients with MDS were divided into *ALOX12* high (*n* = 58) and low groups (*n* = 64) groups, followed by examination of overall survival (OS) using Kaplan–Meier survival analysis. *p* value was calculated using the Gehan-Breslow-Wilcoxon test.

**Figure 4 ijms-25-04583-f004:**
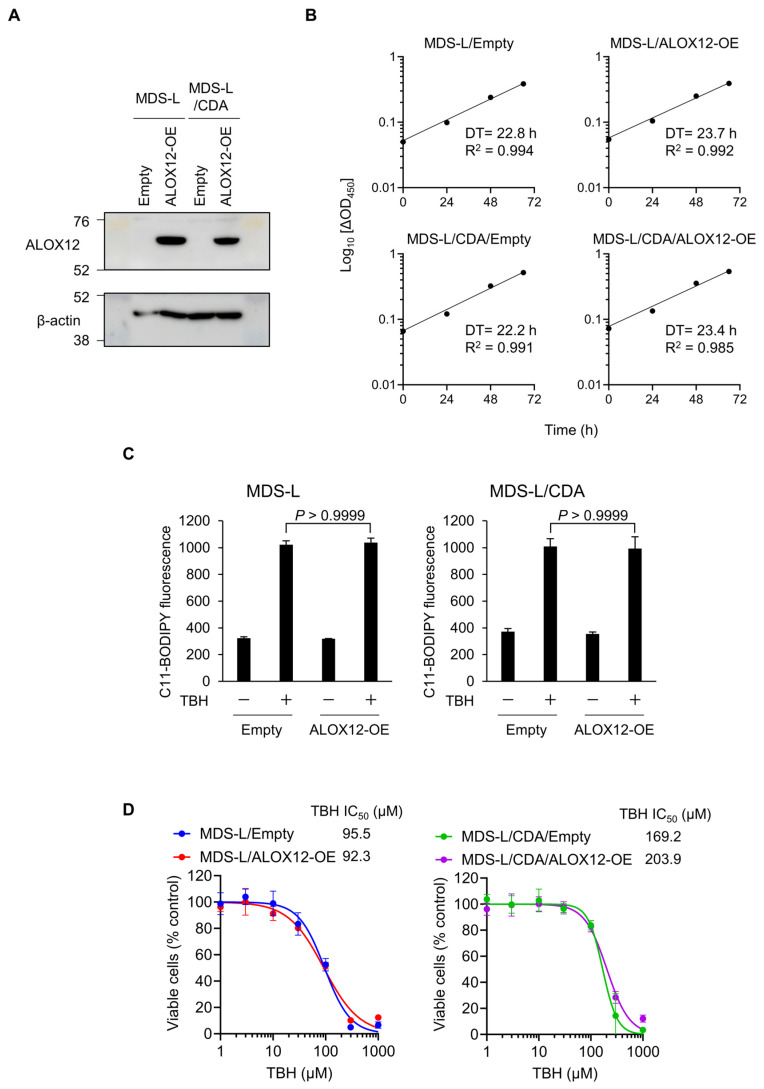
Effects of *ALOX12* gene on cell growth and oxidative stress susceptibility of MDS cells. (**A**) Western blot analysis for MDS-L and MDS-L/CDA cells transfected with empty vector (Empty) and *ALOX12*-expressing vector (*ALOX12*-OE). (**B**) Doubling time of MDS-L and MDS-L/CDA cells overexpressing *ALOX12*. (**C**) Oxidative stress-induced lipid peroxide production in MDS-L and MDS-L/CDA cells overexpressing *ALOX12*. Cells were treated with 200 μM TBH for 2 h and stained with lipid peroxide indicator C11-BODIPY. (**D**) Cytotoxicity of TBH in MDS-L and MDS-L/CDA cells overexpressing *ALOX12* treated with 0–1000 μM TBH for 18 h.

**Figure 5 ijms-25-04583-f005:**
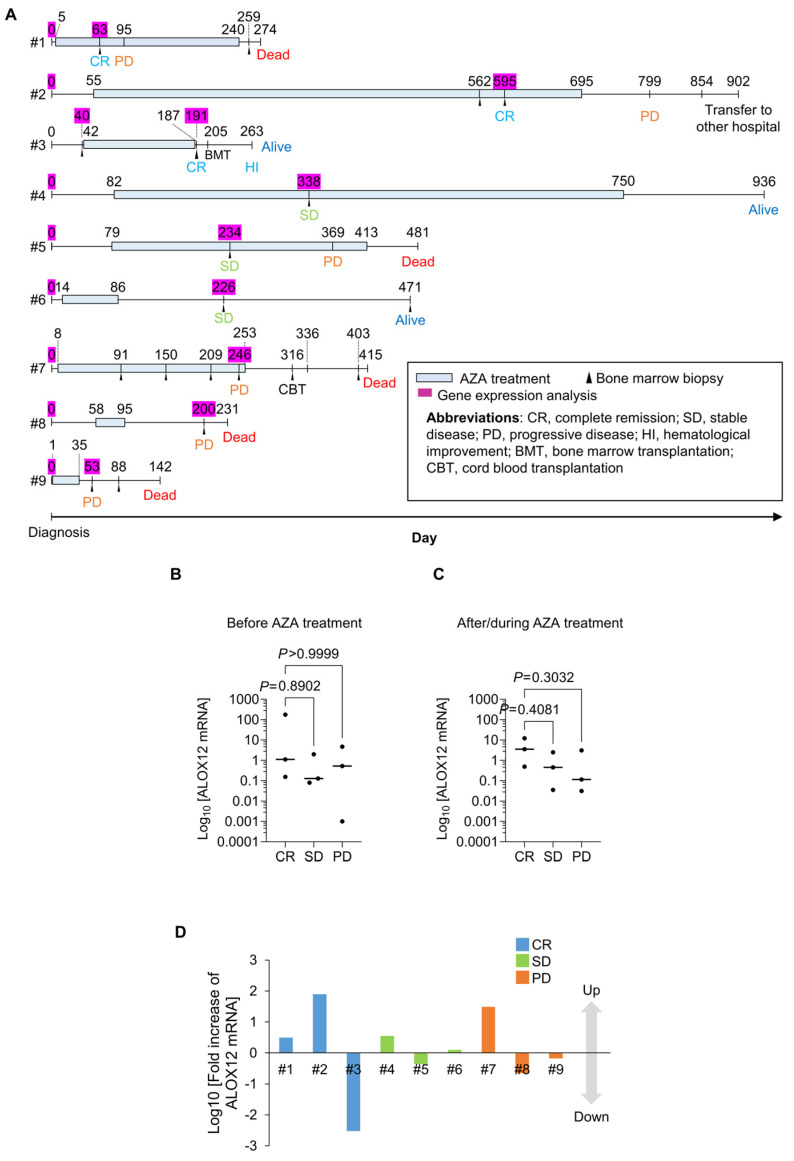
Expression of *ALOX12* genes during treatment with AZA. (**A**) Medical records of patients with MDS enrolled in this study at our hospital. (**B**,**C**) Expression of *ALOX12* gene in bone marrow of nine patients with MDS before (**B**) and after (**C**) treatment with AZA. (**D**) Expression change in *ALOX12* mRNA expression in the bone marrow of nine patients with MDS by AZA treatment.

## Data Availability

We analyzed the publicly available datasets GSE152710 (https://www.ncbi.nlm.nih.gov/geo/query/acc.cgi?acc=GSE152710) accessed on 25 March 2024, GSE145733 (https://www.ncbi.nlm.nih.gov/geo/query/acc.cgi?acc=GSE145733) accessed on 25 March 2024, and GSE58831 (https://www.ncbi.nlm.nih.gov/geo/query/acc.cgi?acc=GSE58831) accessed on 25 March 2024. In addition, materials described in the manuscript, including the relevant raw data, will be available to researchers wishing to use them for non-commercial purposes without breaching participant confidentiality. Microarray data are available at Gene Expression Omnibus (GEO) under accession number GSE233764.

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
