# Peer review of "Enhanced ALOX12 Gene Expression Predicts Therapeutic Susceptibility to 5-Azacytidine in Patients with Myelodysplastic Syndromes"

_ijms, 2024, doi:10.3390/ijms25094583_

Round 1
Reviewer 1 Report
Comments and Suggestions for Authors
The study investigates how ALOX12 gene expression changes upon 5-Aza treatment in MDS cells and whether ALOX12 expression level can suggest survival and prognosis to 5-Aza treatment for MDS patients. Minor comments are listed below:
(1) 'U' and 'M' in Figure 2B need to be annotated in figure legend. If certain primer designing software was used for methylation-specific PCR, please specify.
(2) Please specify the overexpression system used for ALOX12. What vector, tagged or not? It looks like an anti-ALOX12 antibody was used for WB. Why is there no endogenous ALOX12 expression in the empty vector control of those two cell lines?
(3) "There were similar rates of lipid peroxide production in the absence and presence of TBH between ......" please consider rephrasing this sentence as it looks a little confusing.
(4) It is preferred to end the last result section by saying something like "More enrollment is required ......". The limited sample number and variation make it impossible to conclude the correlation, not even seemingly.
Reviewer 2 Report
Comments and Suggestions for Authors
Matasumoto and coworkers analyzed AZA-sensitive genes in MDS to identify markers predicting therapeutic success using AZA. The study deals with an interesting and significant question and is well performed. The text is clearly written. However, the results did not conclusively show that ALOX12 is such a marker. Additional examinations are required as the authors state. Nevertheless, the study is worthy of being published in its actual status.
The authors have to complement several sources of their used drugs and material. As such, the source of the used cell lines is missing.
